# From Traditional to Smart: Exploring the Effects of Smart Agriculture on Green Production Technology Diversity in Family Farms

**Yixin Hu, Mansoor Ahmed Koondhar and Rong Kong \***

College of Economics and Management, Northwest A&F University, Xianyang 712100, China;
huyx0706@nwafu.edu.cn (Y.H.); m_koondhar@nwafu.edu.cn (M.A.K.)
* Correspondence: kr1996@nwafu.edu.cn

**Abstract:** The application of smart agriculture is increasingly becoming a critical force in transforming the traditional methods of agricultural production in China. This change, based on technological innovation, is essential to promoting a sustainable production system in family farms. This study is based on the resource orchestration theory to investigate how smart agriculture affects the diversity of green production technologies (GPTs) on family farms. Based on a sample of 563 family farms surveyed in 2022, this study utilizes propensity score matching (PSM) methods and instrumental variables to analyze the effect of smart-agriculture adoption on the diversity of GPTs on farms. The findings reveal that smart agriculture has significantly increased the diversity of GPTs on farms by 8.5%. Network consulting services, value-added products, and environmental monitoring services are potential impact mechanisms underlying the positive effects of smart agriculture on the diversity of GPTs on farms. Furthermore, the increased diversity of GPTs is more significant on purely plantation farms, farms without contract farming, and farms with high levels of mechanization.

**Keywords:** smart agriculture; green production technology; diversity; family farm; China

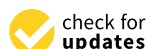

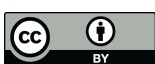

## 1. Introduction

The decline of the ecological environment has become a global issue that has gained international attention in recent years [1]. The unreasonable development of agriculture has caused a series of key issues, such as the overuse of agricultural resources and ecological degradation [2]. Green production technologies (GPTs) are recognized for their role in promoting sustainable agricultural production [3]. Although green production technologies (GPTs) have been adopted worldwide, there is a significant disparity between developed and developing nations. China, as the world's largest developing country, has taken significant steps toward encouraging the implementation of green production technologies (GPTs). Nonetheless, it remains challenging to swiftly transition from its prior conventional agricultural production methods. According to the Ministry of Agriculture's statistics, China's usage of chemical fertilizers accounts for roughly one-third (1.4 million tons) of the world's total annual consumption, exceeding developed nations' usage by 2.5 to 5 times [4]. That is far from green production standards. To balance food security and the development of the ecological environment, there is considerable potential for the widespread adoption of green production technologies (GPTs).

Gao et al. [5] explored the ability of GPTs to improve the quality of arable land and significantly contribute to the transition to green agricultural production, which would entail considerable environmental and economic benefits. Regardless of the significant benefits, Bukchin and Kerret found that the adoption of green production technologies remains low among farmers in developing countries [6]. Family farms, as the mainstay of agricultural production, adopting green production technologies will greatly determine the level of development of green agriculture in China. The intensive and specialized

production characteristics of family farms have enabled them to overcome some of the obstacles to the adoption of green production techniques. Therefore, it is vital to find a green transition path based on family farm characteristics.

In recent years, concepts such as smart farming, smart agriculture, or precision agriculture have become more popular [7]. Sultan et al. [8] think these technologies include climate-smart agriculture (CSA), climate-smart forestry (CSF), climate-smart rangeland management, and climate-smart livestock production. Smart farming is being conducted on a large scale using the IoT, artificial intelligence (AI), and agricultural data analysis in developed nations according to Goel et al. [9]. Mazzetto et al. [10] thought that smart agriculture (SA) is an evolution of precision farming (PF). It is the integration of traditional production and the Internet of Things. The Internet of Things (IoT) technology can link various remote sensors, such as robots, ground sensors, and drones [11]. Sharma et al. [12] explained that this integration has boosted agriculture production due to the high potential to assist farmers. The impulse towards technological advancement has changed traditional agriculture methods and resulted in eco-friendly, sustainable, and efficient farming. Smart agriculture brings numerous benefits to agricultural production through weather monitoring, groundwater detection, and temperature control. IoT-based smart agriculture's common practices include smart farming for sustainable agriculture and smart crop rotations that mitigate issues and challenges related to weeds, plant disease, insects, and other pests according to Zikria et al. [13]. The use of smart agriculture on farms is gradually spreading. This technological breakthrough provides scope for the promotion of green production techniques. Moreover, the cost of using smart agriculture can be evenly spread through large-scale production in the family farm, decreasing the difficulty of adopting smart agriculture and GPT. The above research demonstrates the link between smart agriculture and green production.

Based on Sirmon et al.'s [14] resource orchestration theory, the adoption of smart agriculture on farms will go through three stages of resource integration. First, with the adoption of smart agriculture, farmers will reorganize all resources and solve some traditional agricultural problems, such as water shortages, cost management, and productivity issues according to Farooq et al. [15]. Second, farmers will update their capability by learning new skills; for example, green production knowledge will be attained by learning to operate equipment, such as the application of machines in pest and plant disease identification, robot navigation [16], and wireless underground sensor networks for soil monitoring [17]. Last but not least, there is a combination of resources to create new value and influence farmers' behavior. According to Schukat and Heise [18], farms that use smart farming are more economically productive. It could help farms have more funds to update and adopt GPTs. Farmers can, therefore, combine a diverse range of green production techniques, depending on the mix of resources, to achieve technological efficiency and value transformation and upgrading.

We attempt to verify the pathway of smart agriculture's impact on the diversity of green production. Based on the research base of scholars, we believe there are three possible paths: First, smart agriculture relies on the development of the Internet of Things to provide farmers with a range of web-based advisory services [19]. This makes it easier for farmers to access professional farming technology via the internet, thus enhancing their knowledge of green technology [20,21]. Second, based on internet food safety traceability, smart agriculture increases the added value of agricultural products, which in turn influences farmers' decisions on technological improvements [22]. Third, smart agriculture enables accurate monitoring and assessment of the soil and growing environment. Smart fertilizer management techniques leverage data, sensors, and advanced tools to ensure accurate fertilization levels in agricultural production [23], which builds a proper environment and reduces the difficulty of GPT adoption. It can also ensure the quality of agricultural products [24], which could bring more profits. The expansion of information sources has also influenced farmers' awareness of green production. The dual influence of awareness

and technical knowledge promotes greater adoption of green production techniques among farmers. The mechanisms of influence are shown in Figure 1.

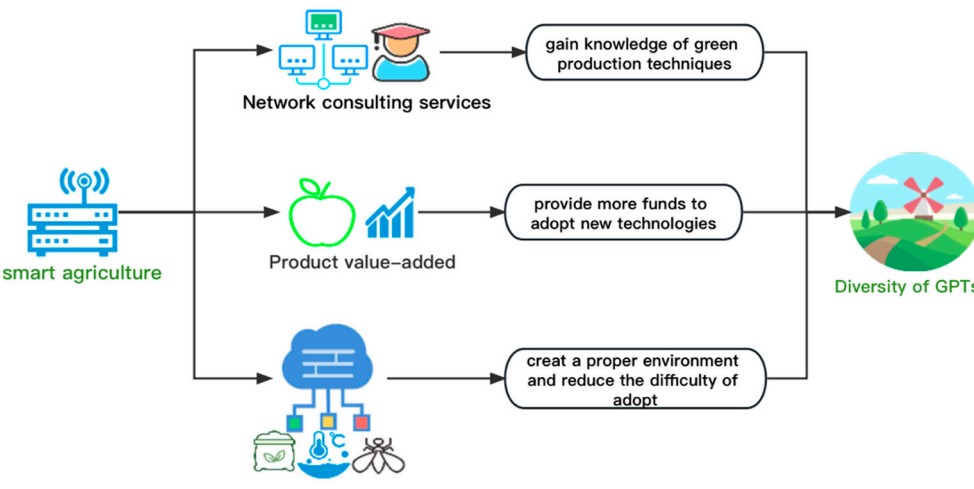

**Figure 1.** Mechanisms of influence.

Current research certainly demonstrates the possible links between smart farming and green production technologies but rests more on the impact of smart farming on a single technology, such as pest control and precision fertilization techniques [23,25]. Secondly, more studies prefer to explore the application of smart agriculture equipment, and fewer study the impact of equipment used on producer behavior. Third, in terms of green production technology adoption, research highlights the influencing factors of adoption [26]. Finally, this study focuses on the implementation of smart agriculture and its impact on the diversity of GPTs on family farms, with a view to providing useful recommendations for sustainable agricultural development in China and other developing countries. The literature on the impacts of smart agriculture on green production provided numerous valuable studies for this paper. However, whether benefits of smart agriculture can encourage farmers to adopt more GPTs is still unclear. Therefore, according to the resource supply path of smart agriculture, this paper explores the impact of smart agriculture on green-production technology through three channels: environmental monitoring, rising value-added product, and network technology services.

The paper is structured as follows. Section 2 presents the data and methodology employed in this study. In Section 3, we analyze the findings and address endogeneity concerns using endogenous transformation models and instrumental variables. Section 4 is the discussion and future research. Finally, the last section offers policy implications and concludes the study.

## 2. Materials and Methods

### 2.1. Data

This paper's research samples were family farms registered by the Ministry of Agriculture and Rural Development. In May 2022, a project team collected data from Shaanxi Province, a primarily agricultural province in Northwest China. The team randomly selected family farms from nine municipalities, covering the topography of Shaanxi Province from north to south. Shaanxi Province, as a largely agricultural province in western China, has significantly different natural conditions for agricultural production, rich policies for cultivating family farms, and many agriculture-related universities and agricultural enterprises, which can better reflect the impact of smart agriculture on farmers' green production behavior under differentiated farm characteristics. The sample research sites are divided into family farm demonstration counties and general counties and districts. Figure 2 shows the specific sample distribution. The project team distributed questionnaires to 650 family

farms, of which 624 were recovered as valid. This indicates an effective recovery rate of 96%. The questionnaire was a comprehensive and systematic interview survey that covered individual farmer characteristics, farm production and operation, financial lending, risk management, and green agriculture. To focus on the adoption of diversity in green production technology, we excluded purely breeding family farms from the sample. As a result, our sample contained 563 family farms.

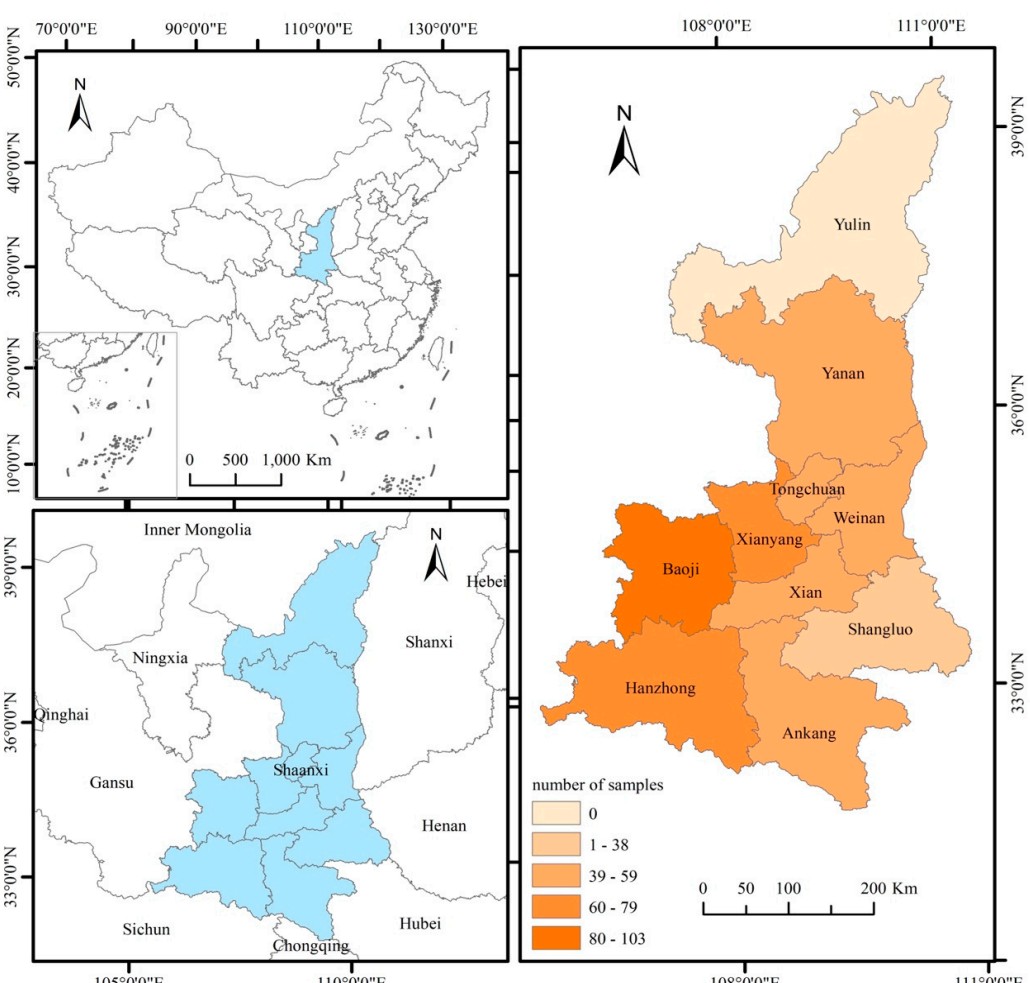

**Figure 2.** Sample distribution.

## 2.2. The Propensity Score Matching (PSM) Model

Based on the current state of development of smart agriculture in China, we define the use of smart agriculture in this paper based on the adoption of at least one smart agricultural device, including smart detection and temperature control devices, smart irrigation systems and drones, etc. The specific equipment is shown in Figure A1. Whether smart farming was adopted served as a binary variable, defined as $C = 1$ if the farm used smart farming and $C = 0$ if it did not. Logit models were then used to estimate fitted values for the conditional probability of participation in smart agriculture, which is an expression of the propensity score (Equation (1)).

$$PS = \Pr(C = 1|X) = E(C = 0|X) \tag{1}$$

where *PS* is the propensity score, and $C = 1$ and $C = 0$ refer to family farms that have or have not adopted smart agriculture, respectively. *X* refers to the observable personal characteristics of the family farmer and the characteristics of the farm. To ensure the matching results' robustness, we selected four matching methods., namely k-nearest neighbor matching

(1 to 4), kernel matching, partial linear matching, and radius matching. The impact of the adoption of smart agriculture on the diversity of green production technology was evaluated through the average treatment effect (*ATT*), defined in Equation (2). In addition to employing PSM based on observable covariates, we conducted a bounds analysis to assess the impact of unobservable factors on the estimated effects.

$$ATT = E(M_1|C = 1) - E(M_0|C = 1) = E(M_1 - M_0|C = 1) \qquad (2)$$

*2.3. Endogenous Switching Regression (ESR)*

While PSM can only address sample selection bias caused by observable factors, Rosenbaum's bounds offer an estimate of how unobservable factors may affect the point estimates [27]. To avoid potential bias from both observable and unobservable factors, such as farmers' risk appetite and usage habits, we use an endogenous conversion model. The ESR model included a selection equation for whether to adopt smart farming and two decision equations for the diversity of green production technologies.

Equation (3) for choosing whether to adopt smart farming on a farm is as follows:

$$A_i^* = \phi(Z_i) + \mu_i \qquad (3)$$

If $A_i^* > 0$, the value of $A_i$ = 1; otherwise, it is 0. Where $A_i$ denotes the latent variable that determines whether farm *i* practices smart farming, $A_i$ = 1 indicates farm *i* is using smart farming equipment; otherwise, it is 0. $Z_i$ refers to the vector of explanatory variables that influence whether a farm practices smart farming and contains 13 control variables and 1 instrumental variable for the distance from the farm to the agribusiness. $\mu_i$ denotes the random error term.

To measure the effect of smart agriculture use on the diversity of green technology adoption in family farms, we constructed the following model of the diversity of green production technologies on farms:

$$Y_i = X_i\beta_i + \delta A_i + \varepsilon_i \qquad (4)$$

In Equation (4), the dependent variable $Y_i$ is the degree of diversity of green production technologies on the farm, $X_i$ is a vector of control variables, $A_i$ denotes the use of smart agriculture on farm *i*, and $\varepsilon_i$ is a random disturbance term. Farms choose whether to use smart agriculture according to their own conditions, and the adoption is affected by certain unobservable factors that are correlated with the outcome variable Y, resulting in a correlation between $A_i$ and $\varepsilon_i$ in Equation (4). Therefore, estimating Equation (4) directly may result in bias due to self-selection issues within the sample. The diversity models of green production technologies corresponding to farms that have or have not used smart agriculture are shown in Equations (5) and (6), respectively.

$$Y_{ia} = X_{ia}\beta_a + \sigma_{ua}\lambda_{ia} + \varepsilon_{ia}, \text{ if } A_i = 1 \qquad (5)$$

$$Y_{in} = X_{in}\beta_n + \sigma_{un}\lambda_{ia} + \varepsilon_{in}, \text{ if } A_i = 0 \qquad (6)$$

$Y_{ia}$ and $Y_{in}$ denote the level of diversity of green production technologies on farms that have or have not used smart agriculture, respectively. $X_{ia}$ and $X_{in}$ denote the factors affecting the diversity of green production technologies on both types of farms, and $\varepsilon_{ia}$ and $\varepsilon_{in}$ are random disturbance terms. To solve the problem of sample-selectivity bias caused by unobservable factors, we introduced both the inverse Mills ratio interest rate and the covariance and the applied the full information great likelihood method to jointly estimate the equations.

We estimated the average effect of smart-agriculture use on the diversity of green production technologies by comparing the expectations of farms with and without the use of smart agriculture under both real and counterfactual hypothetical scenarios. Equation (7) is the diverse expectations of green production technologies for farms using smart agriculture. Equation (8) is the expected value of diversity of green production technologies for farms not using smart agriculture. Equation (9) considers two counterfactual hypothetical scenarios, i.e., the expected value of diversity in green production technologies for farms using smart agriculture in the unused scenario. Equation (10) is the expected value of the diversity of green production technologies on farms not using smart agriculture.

$$E[Y_{ia}|A_i = 1] = X_{ia}\beta_a + \sigma_{ua}\lambda_{ia} \tag{7}$$

$$E[Y_{in}|A_i = 0] = X_{in}\beta_n + \sigma_{un}\lambda_{in} \tag{8}$$

$$E[Y_{in}|A_i = 1] = X_{ia}\beta_n + \sigma_{un}\lambda_{ia} \tag{9}$$

$$E[Y_{ia}|A_i = 0] = X_{in}\beta_a + \sigma_{ua}\lambda_{in} \tag{10}$$

The treatment effect of the diversity of green production techniques on farms using smart agriculture (Equation (11)) was obtained through Equations (7) and (10). $ATU_i$ (Equation (12)) refers to the treatment effect of the diversity of green production technologies on farms not using smart agriculture. We used the mean of $ATT_i$ and $ATU_i$ to estimate the average treatment effect of two types of on-farm smart agriculture use on green production technology diversification.

$$ATT = E[Y_{ia}|A_i = 1] - E[Y_{ia}|A_i = 1] = X_{ia}(\beta_a - \beta_n) + (\sigma_{ua} - \sigma_{un})\lambda_{ia} \tag{11}$$

$$ATU_i = E[Y_{ia}|A_i = 0] - E[Y_{in}|A_i = 0] = X_{in}(\beta_a - \beta_n) + (\sigma_{ua} - \sigma_{un})\lambda_{in} \tag{12}$$

*2.4. Description of Variables*

The diversity of agricultural technology adoption is mostly based on the number of technologies adopted in the production chain. Based on the "Technical Guidelines for Green Development of Agriculture (2018–2030)" released by China's Ministry of Agriculture and Rural Affairs as well as on interviews with agricultural experts, we opted for the following 11 indicators to measure family farms' common green production technologies: deep plowing and loosening, straw return, organic fertilizer application, soil testing and formulation, reduction of chemical fertilizer use, biological pesticides, reduction of pesticide use, agricultural film recycling, agricultural waste recycling, water and fertilizer integration, and water-saving irrigation technology. For each indicator, the green production behaviors could be selected through two choices: (1) the family farm adopted this green production technology and (2) the family farm did not adopt it. The criterion of diversity was defined by the number of total green production behaviors. Accordingly, we measured diversity through the number of green production technologies adopted as a proportion of total green technology in the agricultural production chain. Although we took the impact of smart agriculture on the diverse behavior of green production on farms into consideration, it was also necessary to consider other relevant influences in the control variables to ensure the scientific nature of the research. As such, following the relevant literature [2,28], we selected the family farm owners' characteristics and family farm operational indicators as the control variables. Table 1 presents the descriptive statistics of the data.

As shown in Table 1, 141 of the 565 farms used smart farming, representing 24.95% of the total sample. Table 1 shows the diversity of green technologies on farms, with a mean value of 0.731 for family farms that have adopted smart farming and 0.602 for

those that have not, with a difference at the 1% level of significance. This suggests that the use of smart farming may be a greater incentive to adopt environmentally friendly production techniques. In terms of farm-owner characteristics, younger, more educated, and professional farmers as well as those who have been in the business for a shorter time, seem to be more willing to adopt smart farming, with significant differences in characteristics. In terms of the farms themselves, the smart farming participation group was characterized by larger land areas, higher mechanization values, and higher operating incomes and costs. For other characteristics, participating group farms were more likely to engage in contract farming and to be members of model farms and cooperatives.

**Table 1.** Variable descriptions.

| Variables | Definition | Adopted (A = 141) | Non-Adopted (B = 424) | Mean Difference (A-B) |
|---|---|---|---|---|
| Diversity of GPT adoption | The proportion of AGPT adoption | 0.731 | 0.602 | 0.129 *** [1] |
| Age | Age of the farm's owner | 46.638 | 49.472 | −2.833 *** |
| Education | Years of education of farm's owner | 12.319 | 10.810 | 1.509 *** |
| Professional farmer | 1 if the farm's owner is a professional farmer, 0 otherwise | 0.702 | 0.604 | 0.098 *** |
| Years of planting | Years of participating in agricultural practices | 18.312 | 22.069 | 3.757 *** |
| Land scales | Land size (mu [2]) | 243.455 | 127.419 | 116.026 *** |
| Machine value | Value of total agricultural machine | 10.764 | 9.262 | 1.502 *** |
| Farm income | Farm's total income in 2021 (yuan) | 13.436 | 12.433 | 0.993 *** |
| Farm cost [3] | Farm's total production fees in 2021 (yuan [2]) | 12.472 | 11.505 | 0.967 *** |
| Brand | 1 if the farm registered brand of agricultural products; 0 otherwise. | 0.433 | 0.173 | 0.260 *** |
| Contract farming | 1 if the farm participated in contract farming, 0 otherwise | 0.355 | 0.185 | 0.170 *** |
| Number of labors | The number of owner labor | 2.652 | 2.372 | 0.280 *** |
| Demonstration farms | 1 if the farm is registered as a demonstration farm; 0 otherwise. | 0.766 | 0.521 | 0.245 *** |
| Cooperatives | 1 if the farm participated in cooperatives; 0 otherwise. | 0.546 | 0.384 | 0.162 *** |
| | Instrumental variable | | | |
| Distance | The distance from farm to agribusiness | 1.399 | 1.813 | 0.414 *** |

Note: [1] ***, **, * indicate significance at the 1%, 5%, and 10% levels, respectively. [2] In May 2022, one yuan was equivalent to 0.15 US dollars, and one mu was equivalent to 0.067 hectares. [3] The total cost encompasses expenses related to labor, seeds, fertilizers, pesticides, irrigation water, and other factors.

## 3. Results and Discussion

Using propensity scores derived from a logit model for smart agriculture participation, we performed a PSM estimation and assessed the associated treatment effects on the main outcome variables. To evaluate the impact of unobserved factors on our PSM estimates, we conducted a sensitivity analysis. Furthermore, we verified the robustness of our primary findings through the use of ESR and instrumental variables.

### 3.1. The Determinants of Smart-Agriculture Adoption

This paper estimated a logit model to obtain the propensity scores of farmers' decisions to participate in smart agriculture. The model included household head and household characteristics, which could have influenced farmers' decisions to use smart agriculture and determine which green technologies to adopt. The results of the logit model are presented in Table 2, showing that education, farm cost, brand, and demonstration farms had statistically significant effects on smart-agriculture usage. The influence of educational factors on the adoption of smart farming is consistent with other studies [21,29]. In addition, China rates family farms at different levels, so the indicator of demonstration farms is significant due to the policy effect. Demonstration farms receive higher levels of financial subsidies that will promote the use of smart farming. The marginal effects of these determinants are also reported for better interpretation of their effects.

**Table 2.** The determinants influencing smart-agriculture adoption.

| Variables | (1) Logit Estimates | (2) Marginal Effects |
|---|---|---|
| Age | −0.009 (0.016) | −0.001 (0.002) |
| Education | 0.109 ** (0.043) | 0.017 *** (0.006) |
| Professional farmer | −0.039 (0.246) | −0.006 (0.038) |
| Years of planting | −0.011 (0.012) | −0.002 (0.002) |
| Land scales | 0.000 (0.000) | 0.000 (0.000) |
| Machine value | 0.037 (0.032) | 0.006 (0.005) |
| Farm income | 0.033 (0.087) | 0.005 (0.013) |
| Farm cost | 0.236 ** (0.105) | 0.036 ** (0.016) |
| Brand | 0.852 *** (0.235) | 0.131 *** (0.035) |
| Contract farming | 0.319 (0.251) | 0.049 (0.038) |
| Number of labors | 0.044 (0.056) | 0.006 (0.009) |
| Demonstration farms | 0.600 ** (0.252) | 0.092 ** (0.038) |
| Cooperatives | 0.065 (0.219) | 0.001 (0.034) |
| Constant | −6.23 *** (1.603) | |
| Pseudo R2 | 0.158 | |
| Observations | 563 | |

Note: ***, **, * indicate significance at the 1%, 5%, and 10% levels, respectively.

### 3.2. PSM Estimation for the Effects of Smart Agriculture

3.2.1. Matching Quality

Prior to presenting the treatment estimation results, we assessed the matching quality of both methods by evaluating the overall bias and the common support of the propensity scores. From the results of the balance test (see Tables A1 and A2), it can be seen that, after matching the samples, the standardized bias of the explanatory variables decreased from 38.9% to 2.3–8.5%, and the total bias was significantly lower and less than the 20% level specified by the equilibrium test; the pseudo $R^2$ decreased from 0.157 before settling at 0.002–0.024 after matching; and the LR statistic decreased from 99.61 before matching to 0.87–8.78 after it. Based on the test methods of [21], the results show that the PSM method can effectively reduce the differences in the distribution of explanatory variables between the control and treatment groups and eliminate the estimation bias caused by sample self-selection in our study. Additionally, Figure 3 demonstrates common support in the distribution of the predicted propensity scores between the adoption and non-adoption of smart agriculture. Most observations are within the common range of values (on support in green and red).

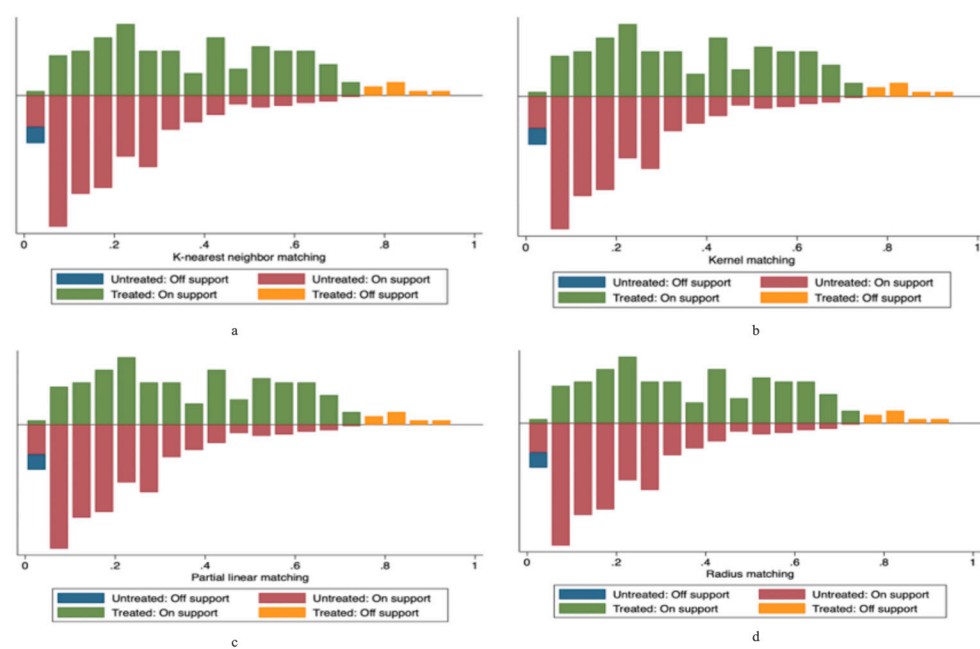

**Figure 3.** Common support of four matching methods.

### 3.2.2. PSM Results

Table 3 shows the average treatment effects of smart agriculture on the diversification of green production technologies on farms. The measurement results were essentially consistent after the use of four different matching methods, all of which were significant at the 1% level, thus indicating that the sample data had good robustness. In addition, the arithmetic mean was selected to characterize the impact effects for later analysis. The study findings indicate a significant and positive impact of smart-agriculture adoption on farms' GPT diversity. Specifically, the implementation of smart agriculture may increase the likelihood of using green production technologies by 8.5% if all farms adopt this approach. The possible reason is that smart agriculture has higher labor efficiency and productivity than conventional agriculture [21], which provides a better practical environment for the adoption of GPT.

**Table 3.** PSM regression results for the effects of smart agriculture on diversity in the GPT.

|  | ATT | S.E. | T-Test |
|---|---|---|---|
| **Panel A: K-nearest neighbor matching** | | | |
| Diversity of GPT | 0.086 *** | 0.025 | 3.44 |
| **Panel B: Kernel matching** | | | |
| Diversity of GPT | 0.085 *** | 0.023 | 3.71 |
| **Panel C: Partial linear matching** | | | |
| Diversity of GPT | 0.082 *** | 0.029 | 2.79 |
| **Panel D: Radius matching** | | | |
| Diversity of GPT | 0.086 *** | 0.025 | 3.48 |
| Mean | | 0.085 | |
| Balancing property satisfied | | YES | |
| Common support imposed | | YES | |
| Number of treated | | 134 | |
| Number of controls | | 411 | |
| Combined | | 545 | |

Note: ***, **, * indicate significance at the 1%, 5%, and 10% levels, respectively.

### 3.2.3. Sensitivity Analysis

As mentioned earlier, the PSM method assumes that the decision to adopt smart agriculture is based solely on observable factors. However, there may be some unobserved factors at play. To test the susceptibility of the estimated treatment effect to these unobserved factors, we conducted a sensitivity analysis using Rosenbaum bounds. The results of this analysis are presented in Table A3. In this method, Γ is a measure of sensitivity to hidden bias. The closer Γ is to 1, the greater the results' sensitivity to possible hidden bias, whereas Γ being closer to 2 reflects a lower sensitivity [30]. As shown in Table A3, four matching methods of the gamma coefficient were still significant at the 10% confidence level after 2. The PSM model was constructed based on the available confounding variables and its estimated propensity values are robust, thus supporting the plausibility of our findings.

### 3.3. Robustness Tests Based on Instrumental Variables

This paper employed the ESR model to verify the robustness of our PSM results and control for potential bias caused by unobserved factors. To ensure model identification, we used the distance to the nearest agribusiness entity as a potential instrumental variable for smart-agriculture adoption, which is commonly used in similar studies [31,32]. A valid instrument should satisfy relevance and exogeneity properties. First, it should be highly correlated with smart-agriculture adoption as only agribusiness entities are likely to offer smart devices to farmers. Moreover, the proximity of a farm to an agribusiness entity leads to a higher spillover effect, increasing the likelihood of nearby farmers adopting smart agriculture. Second, its exogenous nature could be verified by the fixed distance between a farm and the nearest agribusiness entity, even before the farmer decides to adopt smart agriculture. Thus, the distance is less likely to have a direct impact on farmers' GPT diversity. The estimation results are presented in Table A4, and as expected, the distance has a significant and negative effect on smart-agriculture adoption, implying that longer distances tend to reduce the probability of adoption. The Wald F statistics were all significant, indicating the absence of a severe weak instrument problem.

Table 4 presents the estimated treatment effects of smart-agriculture adoption on the diversity of GPT among farmers, which were consistent with the findings obtained from PSM. However, the magnitude of the treatment effect was larger, with an average of 14.2%. These results were further confirmed by the ESR model, indicating that smart-agriculture adoption significantly improves farmers' GPT diversity. It is important to note that the treatment effect from the ESR model should be interpreted as a local average due to the introduction of an additional instrument.

**Table 4.** Treatment effect of smart-agriculture adoption on the diversity of the GPT.

|  | ATT | ATU | ATE |
|---|---|---|---|
| Diversity of GPT | 0.142 *** (0.011) | 0.078 *** (0.006) | 0.094 *** (0.005) |

Note: ***, **, * indicate significance at the 1%, 5%, and 10% levels, respectively.

### 3.4. Channel Analysis

Table 5 presents the estimations of smart agricultural potential channels and their effects on the diversity of green production technologies, using both KNNM and KN methods. The results from both methods were largely consistent, as supported by previous studies [33–36]. We found that smart agriculture has a significant and positive effect on farms' agricultural production, including network technology services, value-added product, and environmental monitoring. More specifically, ATEs suggest that if all farms were to adopt smart agriculture, the network technology services, value-added product, and environmental monitoring could be increased by 17%, 78.7%, and 33.7%, respectively. We thus conclude that improved access to network technology, value-added products, and environmental monitoring services are potential changes through which smart farming has a clear positive impact on the technological diversity of the green production of farms.

**Table 5.** PSM regression results for the effects of smart agriculture on potential channels.

| Variables | ATT | ATU | ATE |
|---|---|---|---|
| **Panel A: K- Nearest neighbor matching (KNNM)** | | | |
| Network technology services | 0.139 ** (0.059) | 0.180 *** (0.055) | 0.170 *** (0.049) |
| Value-added product | 0.784 *** (0.037) | 0.788 *** (0.063) | 0.787 *** (0.053) |
| Environmental monitoring services | 0.326 *** (0.066) | 0.341 *** (0.065) | 0.337 *** (0.057) |
| Balancing property satisfied | | Yes | |
| Common support imposed | | Yes | |
| Observations | | 563 | |
| Number of treated | | 134 | |
| Number of controls | | 411 | |
| Combined | | 545 | |
| **Panel B: Kernel matching (KN)** | | | |
| Network technology services | 0.138 *** (0.048) | 0.185 *** (0.048) | 0.173 *** (0.044) |
| Value-added product | 0.783 *** (0.036) | 0.749 *** (0.054) | 0.757 *** (0.047) |
| Environmental monitoring services | 0.312 *** (0.056) | 0.350 *** (0.057) | 0.341 *** (0.052) |
| Balancing property satisfied | | Yes | |
| Common support imposed | | Yes | |
| Observations | | 563 | |
| Number of treated | | 134 | |
| Number of controls | | 411 | |
| Combined | | 545 | |

Note: ***, **, * indicate significance at the 1%, 5%, and 10% levels, respectively.

### 3.5. Heterogeneity Analysis

To further explore the differential impact of the adoption of smart farming by family farms with different characteristics, we grouped farms according to their type, namely whether they were involved in contract farming and their level of machinery value. The PSM results are shown in Table A5. The level of green production technology diversity enhancement appears to be more significant (with an 11.2% increase) after the adoption of smart agriculture in pure plantation farms. This is likely due to the current focus on smart farming for irrigation, fertilization, and weed control, which is more conducive to improving green farming techniques in planting aspects. This is in line with the aims of smart agriculture regarding food security [37]. Furthermore, farms without contract farming had a greater increase in GPT diversity levels after using smart farming, with an average treatment effect of 7.2%. With contracts in place to ensure farm sales and farmer revenue, farmers are less motivated to pursue additional technical advancements. The insignificant effect of contract farming on technology has also been verified in other studies [38]. Accordingly, the number of farms adopting green production technologies would not be significantly higher. Finally, the average treatment effect for farms with higher levels of mechanization was 9.1%, significant at the 1% level. Possible reasons for this could be that farms with high levels of mechanization have more productive capital and technical knowledge and are more willing to embrace the new technologies brought by smart farming.

## 4. Discussion

The focus of the discussion will be based on the results of the study and will mainly cover the adoption of smart agriculture and GPT, the impact of smart agriculture on green production technologies, and future research directions.

### 4.1. Adoption of Smart Agriculture and GPT

Empirical studies on the adoption of GPT and smart agriculture have proliferated globally, revealing that factors such as human capital and household resources play significant roles [16,18]. Digital finance [39] and government support [26] are also major factors affecting farmers' adoption of GPT and smart agriculture. Against the backdrop of the current global COVID-19 outbreak, some scholars have found that the adoption of green and smart agriculture can effectively address the impact of COVID-19 on agricultural development [40]. Smart farming offers advantages not only to scientists and agronomists but also to farmers, as it enables them to leverage modern technologies and devices that enhance productivity, minimize costs, and improve product quality [7]. Based on the advantages of smart agriculture and green production technology adoption, scholars have further explored the barriers to adoption, mainly including several demographic and socioeconomic factors that may impede farmers' adoption of AGPTs [41,42]. These include individual endowment factors such as age and education level [28], family characteristics such as arable area and farming experience, and cognitive factors such as environmental knowledge and responsibility awareness [43]. There is consistency between these studies and the results of our influencing factor in the logit model.

Additionally, policies such as agricultural subsidies and organizational support may also play a role. For example, family farms in China are graded according to administrative regions and level of farm development, including county model farms, municipal model farms, and provincial model farms, and whether being a model farm has an impact on the farm's access to financial subsidies, which in turn affects whether the farm has sufficient funds to adopt smart agriculture and GPTs. Therefore, whether it is a demonstration farm or not also has an influence on the adoption of smart agriculture.

### 4.2. The Impact of Smart Agriculture on the Diversity of GPTS

According to the results of the ATT on the impact of smart agriculture on green production technology diversity, farms that adopted smart agriculture were able to increase their green production technology adoption percentage by 8.5%. In terms of the counterfactual, farms using smart farming that do not adopt smart farming have a green production diversity participation rate of 64.6% (adoption of 7 green production technologies), but their green technology adoption diversity increases to 73.1% (adoption of 8 green production technologies) due to the use of smart farming, an increase of 8.5% and a growth rate of 13.3%. The results validate the role of technology in contributing to green agricultural development, which is in line with the direction of Beddington's research [44].

In the mechanism test, we found that smart farming has a significant positive impact on agricultural production on farms, including network technology services, value-added products, and environmental monitoring. The results of the ATEs show that network technology services, value-added products, and environmental monitoring could be increased by 17%, 78.7%, and 33.7%, respectively, if all farms adopted smart farming. Sagheer et al. [45] thought that smart agriculture offers the possibility of diversifying into green technologies through detection and control. Romeo et al. also found that smart farming uses integrated services to provide technical support to farmers, thus contributing to product quality [46]. Improved product quality helps to improve farm income and provides financial support for the adoption of green production technologies. Smart agriculture makes automated operations possible, effectively improving the efficiency of environmental monitoring by Ahmed et al. [47]. For example, smart irrigation systems integrate IoT technology with smart agriculture to conserve water consumption during the irrigation of

agricultural land [48]. This not only improves GPT levels but also reduces production costs on the farm.

The impact of agricultural production on environmental pollution also becomes measurable when the level of environmental monitoring is increased. Instead of farmers relying on experience for their farming activities and simply increasing fertilizers and pesticides as the only solution to ensure yields, more scientific production solutions have been created by smart farming systems. Our research findings bring new empirical evidence for the development of green production methods on family farms. Family farms have stronger economic power compared to smallholder farmers and with supportive national policies. It is far more willing to adopt new technologies than smallholder farmers [49]. Based on the impact of smart agriculture on the diversity of GPT, promoting smart agriculture in farms is a good way to accelerate the modernization and greening of Chinese agriculture.

*4.3. Future Research*

Our study aims to investigate the impact of smart-agriculture adoption on the diversity of GPTs. While this objective aligns with prior research, we extend the literature in three key ways. First, we use representative samples on GPT adoption from family-farm surveys in rural China in 2022. Second, we provide fresh evidence of influential mechanisms. Third, we find that the diversity of GPTs is more significant on purely cultivated farms, farms without contract farming, and farms with high levels of mechanization. This facilitates the targeted promotion of smart farming.

The object of this paper is to grow family farms, and further research can be conducted in the future depending on differences in the type of farm (e.g., farm or leisure farm), farm cultivars (types of vegetables), etc. In this way, the hot areas of smart farming applications can be analyzed, which in turn will complement the future directions of agricultural technology improvement. In addition, the adoption of smart agriculture in this paper is based on farmers' individual choices, and the differences in the effects of active adoption and policy support can be further analyzed in light of the future implementation of smart rural projects in China. Finally, a separate discussion can be conducted to analyze the differences in the adoption performance of drones, smart seeding, and smart irrigation equipment based on the type of smart agriculture equipment, deepening the impact of smart agriculture on green production technologies.

**5. Conclusions**

Smart agriculture has infiltrated agricultural production, and the increased adoption of green technologies by technological advances is crucial for developing countries. In this study, we estimated the impact of smart agriculture on GPT diversity through a field survey of 563 family farms in Shaanxi Province, one of China's agricultural production provinces. PSM and ESR were applied to address possible selection bias from observable and unobservable factors. We found that the use of smart agriculture had a significant positive impact on GPT diversity, with an increase of 8.5%. Smart agriculture influences the technological diversity of green production on farms through web-based technical services, value-added products, and environmental monitoring services. Furthermore, GPT diversity increased more significantly (by 11.2%) after the adoption of smart agriculture in pure plantation farms. Farms without contracts were more strongly affected by the green incentive effect of smart farming, with a 7.2% improvement. Finally, higher levels of mechanization resulted in a greater diversity of GPTs on farms that adopted smart agriculture, with a significant increase of 9.1%.

The policy implications of our findings are noteworthy. Although the widespread adoption of smart agriculture and GPTs is expected to transform the agricultural and environmental sectors, it is imperative to adopt a more inclusive development strategy. More support should be provided to facilitate the adoption of smart agriculture. First, smart agriculture software companies could simplify their equipment operations, taking intelligence and convenience as the principles of smart agriculture manufacturing. It

can customize its services according to the type of farm. For planting farms, testing indicators can be matched to planting varieties, and intelligent agricultural systems such as the planting encyclopedia can be developed to spread knowledge of green planting more comprehensively. This method may lower the threshold for the use of smart agriculture and make software in line with the knowledge level of farmers in developing countries. Second, government departments could provide farms with financial subsidies to reduce the cost of smart-agriculture adoption. Empirical experience shows that the contribution of smart farming to green production is significant, controlling for other operational characteristics of the farm. Reducing the cost of using smart farming is, therefore, an important way to promote modernization and greening of agricultural development. Particular attention should be given to the smaller and lower mechanization farms in the less developed regions. Third, the government could expand the construction of rural infrastructure network facilities to break the hardware barriers to smart agriculture applications and lay the foundation for subsequent smart agriculture upgrades.

As the adoption of smart agriculture can be seen as a dynamic game between farmers and producers in terms of costs and revenues, future research can focus on the dynamic impact of smart agriculture on environmentally sustainable behaviors, exploring the long-term impacts of smart agriculture, and its relationship with agricultural yields and green behaviors. However, caution should be exercised when interpreting the results of our study. More research is needed to analyze the heterogeneity of adoption costs, as the costs and outputs of smart farming adoption may vary considerably across different smart farming practices, thus influencing farmers' adoption of GPTs. It would also be useful to further explore the relationship between smart farming and farmers from the perspective of behavioral economics, such as the moderating role of farmers' risk perceptions and preferences.

**Author Contributions:** Conceptualization, Y.H.; methodology, Y.H.; data curation, Y.H.; writing— original draft preparation, Y.H.; writing—review and editing, Y.H., M.A.K., and R.K.; visualization, Y.H.; supervision, R.K.; funding acquisition, R.K. All authors have read and agreed to the published version of the manuscript.

**Funding:** This research was funded by the National Natural Science Foundation of China, grant number 72273107 and 71773094, Northwest Agriculture and Forestry University Major Incubation Program, grant number 2452021168.

**Institutional Review Board Statement:** Not applicable.

**Informed Consent Statement:** Not applicable.

**Data Availability Statement:** The data presented in this study are available upon request from the corresponding author.

**Conflicts of Interest:** The authors declare no conflict of interest.

## Appendix A

This appendix contains five tables and one figure that display information on PSM match quality, sensitivity analysis, ESR regression results, heterogeneity analysis, and smart equipment.

More specific equipment for smart agriculture, including smart sowing, smart spraying, and smart plowing equipment. Machines controlled by the network, belonging to the collection and operation desk for real-time monitoring and analysis, are shown in Figure A1.

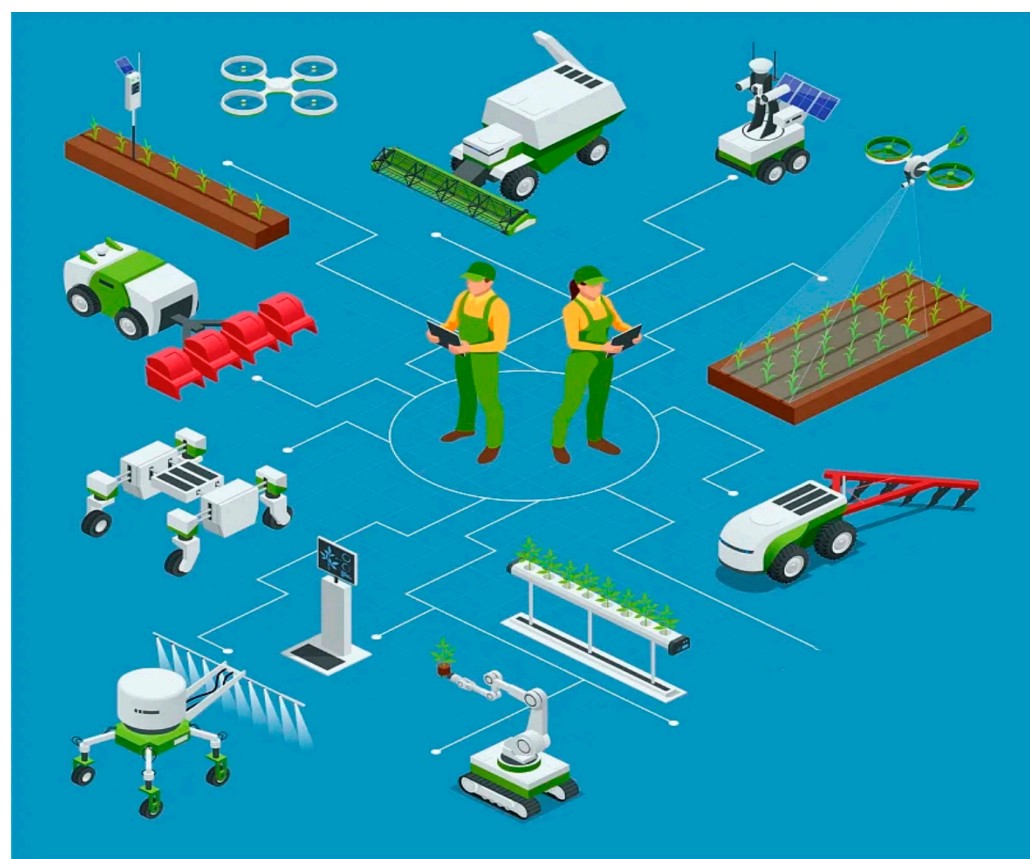

**Figure A1.** Smart agriculture equipment.

**Table A1.** Overall PSM quality indicators before and after matching.

| Method | Sample | Ps R2 | LR chi2 | Mean Bias (%) |
|---|---|---|---|---|
| | Unmatched | 0.157 | 99.61 | 38.9 |
| K Nearest Neighbor Matching | Matched | 0.006 | 2.08 | 3.1 |
| Kernel matching | Matched | 0.002 | 0.87 | 2.3 |
| Partial linear matching | Matched | 0.024 | 8.78 | 8.5 |
| Radius matching | Matched | 0.005 | 1.84 | 2.7 |

**Table A2.** Covariate balance test.

| Variables | Unmatched Matched | Mean Treated | Control | %Bias | T Value | $p >$ |t| |
|---|---|---|---|---|---|---|
| Age | U | 46.638 | 49.472 | −31.7 | −3.24 | 0.001 |
| | M | 47.09 | 46.688 | 4.5 | 0.36 | 0.717 |
| Education | U | 12.319 | 10.81 | 54.4 | 5.56 | 0.000 |
| | M | 12.157 | 12.062 | 3.4 | 0.30 | 0.764 |
| Professional farmer | U | 0.702 | 0.604 | 20.6 | 2.09 | 0.037 |
| | M | 0.701 | 0.716 | −3.1 | −0.27 | 0.789 |
| Years of planting | U | 18.312 | 22.069 | −31.9 | −3.23 | 0.001 |
| | M | 18.776 | 19.629 | −7.3 | −0.62 | 0.538 |
| Land scales | U | 243.45 | 127.42 | 36.3 | 4.44 | 0.000 |
| | M | 201.01 | 210.79 | −3.1 | −0.24 | 0.811 |
| Machine value | U | 10.764 | 9.262 | 41.4 | 3.98 | 0.000 |
| | M | 10.718 | 10.66 | 1.6 | 0.15 | 0.877 |

**Table A2.** *Cont.*

| Variables | Unmatched | Mean | | %Bias | T Value | $p > |t|$ |
| | Matched | Treated | Control | | | |
|---|---|---|---|---|---|---|
| Farm income | U | 13.436 | 12.443 | 47.1 | 4.84 | 0.000 |
| | M | 13.333 | 13.285 | 2.2 | 0.21 | 0.834 |
| Farm cost | U | 12.472 | 11.505 | 66.4 | 6.95 | 0.000 |
| | M | 12.366 | 12.338 | 1.9 | 0.16 | 0.875 |
| Brand | U | 0.433 | 0.173 | 58.8 | 6.49 | 0.000 |
| | M | 0.403 | 0.418 | −3.4 | −0.25 | 0.805 |
| Contract farming | U | 0.355 | 0.185 | 38.9 | 4.22 | 0.000 |
| | M | 0.328 | 0.339 | −2.6 | −0.19 | 0.847 |
| Number of labors | U | 2.6525 | 2.372 | 17.3 | 1.90 | 0.057 |
| | M | 2.5896 | 2.639 | −3.1 | −0.26 | 0.798 |
| Demonstration farms | U | 0.766 | 0.521 | 52.7 | 5.21 | 0.000 |
| | M | 0.754 | 0.757 | −0.8 | −0.07 | 0.944 |
| Cooperatives | U | 0.546 | 0.384 | 32.9 | 3.40 | 0.001 |
| | M | 0.537 | 0.532 | 1.1 | 0.09 | 0.927 |

**Table A3.** Rosenbaum bounds sensitivity analysis.

| Gamma | KNNM | | KM | | PLM | | RM | |
| | Sig+ | Sig− | Sig+ | Sig− | Sig+ | Sig− | Sig+ | Sig− |
|---|---|---|---|---|---|---|---|---|
| 1 | 0.000 | 0.000 | 0.000 | 0.000 | 0.000 | 0.000 | 0.000 | 0.000 |
| 1.1 | 0.000 | 0.000 | 0.000 | 0.000 | 0.000 | 0.000 | 0.000 | 0.000 |
| 1.2 | 0.000 | 0.000 | 0.000 | 0.000 | 0.000 | 0.000 | 0.000 | 0.000 |
| 1.3 | 0.000 | 0.000 | 0.000 | 0.000 | 0.000 | 0.000 | 0.000 | 0.000 |
| 1.4 | 0.000 | 0.000 | 0.000 | 0.000 | 0.000 | 0.000 | 0.000 | 0.000 |
| 1.5 | 0.002 | 0.000 | 0.001 | 0.000 | 0.002 | 0.000 | 0.002 | 0.000 |
| 1.6 | 0.005 | 0.000 | 0.004 | 0.000 | 0.004 | 0.000 | 0.005 | 0.000 |
| 1.7 | 0.012 | 0.000 | 0.009 | 0.000 | 0.010 | 0.000 | 0.012 | 0.000 |
| 1.8 | 0.023 | 0.000 | 0.018 | 0.000 | 0.020 | 0.000 | 0.023 | 0.000 |
| 1.9 | 0.040 | 0.000 | 0.032 | 0.000 | 0.036 | 0.000 | 0.040 | 0.000 |
| 2 | 0.065 | 0.000 | 0.053 | 0.000 | 0.058 | 0.000 | 0.064 | 0.000 |

**Table A4.** Estimation of smart-agriculture adoption on the diversity of GPT.

| Variables | (1) Select Equation | (2) Adoption | (3) Nonadoption |
|---|---|---|---|
| Constant | −2.854 *** | 0.622 ** | 0.177 |
| | (0.861) | (0.280) | (0.130) |
| Age | −0.008 | −0.004 | −0.000 |
| | (0.010) | (0.002) | (0.001) |
| Education | 0.059 ** | 0.003 | 0.009 ** |
| | (0.025) | (0.006) | (0.004) |
| Professional farmer | −0.052 | 0.025 | 0.065 *** |
| | (0.143) | (0.034) | (0.021) |
| Years of planting | −0.006 | 0.001 | −0.001 |
| | (0.007) | (0.002) | (0.001) |
| Land scales | 0.000 | 0.000 | −0.000 |
| | (0.000) | (0.000) | (0.000) |
| Machine value | 0.019 | 0.016 *** | 0.008 *** |
| | (0.019) | (0.005) | (0.002) |
| Farm income | 0.001 | 0.020 ** | 0.004 |
| | (0.037) | (0.008) | (0.005) |
| Farm cost | 0.142 *** | −0.023 * | 0.016 * |
| | (0.054) | (0.013) | (0.009) |

**Table A4.** *Cont.*

| Variables | (1) Select Equation | (2) Adoption | (3) Nonadoption |
|---|---|---|---|
| Brand | 0.473 *** | 0.021 | −0.045 |
| | (0.145) | (0.038) | (0.028) |
| Contract farming | 0.200 | 0.030 | 0.044 * |
| | 0.152 | (0.034) | (0.026) |
| Number of labors | 0.021 | 0.004 | 0.004 |
| | 0.039 | (0.008) | (0.007) |
| Demonstration farms | 0.313 ** | −0.023 | −0.047 ** |
| | 0.145 | (0.013) | (0.021) |
| Cooperatives | 0.010 | 0.049 | 0.041 ** |
| | 0.135 | (0.032) | (0.020) |
| Distance | −0.203 *** | | |
| | (0.070) | | |
| Rho1 | | −0.013 | |
| | | (0.381) | |
| Lns1 | | −1.783 *** | |
| | | (0.060) | |
| Rho0 | | | −0.221 |
| | | | (0.230) |
| Lns0 | | | −1.64 *** |
| | | | (0.039) |
| Likelihood | −155.342 | | |
| Wald test | 61.47 *** | | |
| Observations | 563 | | |

Note: ***, **, * indicate significance at the 1%, 5%, and 10% levels, respectively.

**Table A5.** PSM regression results for heterogeneity analysis.

| Variable | ATT | | | | | |
|---|---|---|---|---|---|---|
| | Type of Farm | | Contract Farm | | Machine Value | |
| | =plantation farm | =Combined farm | =1 | =0 | >average | <average |
| Diversity of GPT | 0.112 *** | 0.067 | 0.052 | 0.072 ** | 0.091 *** | 0.076 |
| | (0.030) | (0.043) | (0.041) | (0.030) | (0.027) | (0.056) |

Note: ***, **, * indicate significance at the 1%, 5%, and 10% levels, respectively.

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
