# Peer review of "From Traditional to Smart: Exploring the Effects of Smart Agriculture on Green Production Technology Diversity in Family Farms"

_agriculture, doi:10.3390/agriculture13061236_

Round 1

Reviewer 1 Report

The article is devoted to the study of the impact of smart agriculture on the development of production technologies on family farms.

1) In general, the article has a scientific approach, but it is more like a statistical report. We need more real examples, practical sense.

2) What do the obtained statistical characteristics give? We also need more graphs, diagrams, correlations.

3) It is also necessary to add an analysis in the context of farming sectors (vegetables, livestock, agriculture, etc.).

4) An analysis is also needed in the context of the type of smart technologies used. Please specify what kind of smart technologies are being used - robotic harvesting, soil monitoring, remote monitoring of fields.

5) Add pictures illustrating points 3 and 4. This will increase the attractiveness of the article for readers.

6) Fig. 2 - explain in more detail what is shown by the shades of color? Add it to the description of the picture

7) Fig. 3-6. Write explanations for these figures from a practical point of view. What do all 4 figures tell the reader? I also recommend numbering this drawing with one number, for example: 3 (a, b, c, d).

8) In the literature review, it is desirable to write the name of the researcher, year. For example: In the work of Irish researchers Y. Kaliani and R. Collier (2021) [9], it was found…..

9) It is also not clear what is the diversity of GPT, to which the article is devoted. How do the numbers in the results show diversity? How does it practically manifest itself in farms? More specifics of practical sense are needed!

Reviewer 2 Report

The authors provide an interesting study on the impacts of smart agriculture on the diversity of Green Production Technologies (GPT) adopted on family farms in a study region located in rural China. The manuscript has publication potential but the following items should be addressed in order to strengthen its contribution to scholarship.

First, the authors need to clearly state and defend the purpose of their research. The closest they come to doing this right now is the following sentence that starts on line 90: "...this study focuses on the implementation of smart agriculture and its impact on the diversity of GPTs on family farms, with a view to providing useful recommendations for sustainable agricultural development in China and other developing countries." Why is this study needed? Who needs to know how smart agriculture impacts the diversity of GPTs? Does the existing literature argue that this gap needs to be filled? If so, why? If not, why is it helpful to fill this gap? What can we learn from this? This needs to be explained in the introduction where the authors state their research purpose.

Second, the authors need to more clearly define their key terms. What exactly is smart agriculture? What exactly counts as a GPT? What's the difference between smart agriculture and GPT? Why would one impact the other and why should we assume the impact only goes in one direction and not both directions or the other way around (essentially, what is causing what and how in this situation)? If the purpose of the study is to learn how smart agriculture impacts GPT diversity then these things need to be defined and explained more clearly in the introduction.

Third, the literature review is very brief. Much of the actual discussion of the literature takes place in the introduction rather than in the literature review section where it belongs. I recommend moving this material to its proper place so we can better see the gap in the literature the study is hoping to fill and the reason why this gap is worth filling. Moreover, the authors should consider whether or not it is possible to derive testable hypotheses from what is already known in the existing literature. This would make the paper seem less like it was fishing for statistically significant coefficients and more like it was designed to test theoretical propositions. How exactly do the authors expect the world to work? Does your model confirm or overturn this expectation and what did you learn from this?

Fourth, the authors should provide some discussion of the questions used to obtain the data analyzed. What questions were asked, why, and how? This is very unclear at the moment so it's really hard to understand how to interpret the analysis results. Question wording matters. Not seeing how the questions were asked makes it impossible to know for sure whether or not the authors are properly interpreting their findings.

Fifith, the authors should justify the choice of their study region. Why does it makes sense to study this particular region? How does this region make it possible for you to infer your findings to other similar regions and to what extent are you limited in your ability to do so? The authors should give the reader some understanding of the scope to which their findings apply.

Finally, the authors should provide a stronger explanation of the significance of their findings in the discussion section. In particular, the authors need to speak more directly to their stated study purpose. If the purpose of the study is to investigate the impacts of smart agriculture on the diversity of GPT adoption, what exactly did you learn about this and how is this relevant to policy-making? This is unclear. Furthermore, if all we learn is that one impacts the other but we don't really know how, then what exactly have we learned? For example, if we know that the adoption of smart agriculture encourages the adoption of more diverse GPTs then what can we do with this information if we have no idea why families adopt smart agriculture in the first place? What levers do you recommend policy-makers pull if the goal is to increase the adoption of more diverse GPTs? This is unclear but it seems to be quite critical to the contribution this study is hoping to make.

Reviewer 3 Report

This is a very interesting and carefully prepared study. Of course, it can be doubted whether such sophisticated analysis can be carried out based on data obtained through surveys - I trust the authors that it can (the quality of the collected data is crucial, but it is often difficult to achieve this using surveys). In this context, it is difficult to understand how it was verified whether a farmer uses "smart farming" or not - in my opinion, this should be exposed because it is the basis of the logit model (I did not understand it). The methodology used is very complex, and the whole process of data processing is quite complicated - to make it easier for readers to understand the meaning of the various stages, I suggest preparing and posting a graphical framework (in the methodology section). Some dissatisfaction may be caused by the scope of the discussion carried out - I suggest a deeper reference to the literature on this point.

Reviewer 4 Report

This study investigates how smart agriculture influences the diversity of environmentally friendly production technologies on family farms. This study uses propensity score matching (PSM) methods and instrumental variables to analyze the effect of smart agriculture adoption on farm GPT diversity.   The findings reveal that smart agriculture has significantly increased the diversity of GPTs on farms.

Authors are invited to summarize their contributions as a list of short sentences in the introduction.

The proposed hypotheses are important and justified. Experimental and analytical methodologies are appropriate and feasible.

Further details regarding the reproducibility of the proposed experimental procedures and analysis are to be provided.

There is enough paper neutral hypothesis tests, including positive controls and quality checks.

Round 2

Reviewer 1 Report

However, in my opinion, attention should be paid to the following remarks, which have not been corrected by the authors:

1) Add the names of researchers who deal with this issue. This will show respect for the researchers you cite.

2) Fig. 4-6 combined into one drawing, signing a, b, c (however, I leave this remark at the discretion of the editors).

3) I consider it necessary to add a graphical interpretation of the obtained statistical results, which will be more clearly perceived by the reader

4) From the authors' response: “Point 4: An analysis is also needed in the context of the type of smart technologies used. Please specify what kind of smart technologies are being used - robotic harvesting, soil monitoring, remote monitoring of fields.

Response 4: Thank you very much for your suggestion. Based on the current state of development of smart agriculture in China, we define the use of smart agriculture in this paper based on the adoption of at least one smart agricultural device, including smart detection and temperature control devices, smart irrigation systems and drones, etc . This part is in line 414.”

I did not see in line 414 (in the new and in the old versions) what this remark is about. Please describe in more detail in the article the practical use of intelligent agricultural devices.

5) From the authors' response: “Point 5: Add pictures illustrating points 3 and 4. This will increase the attractiveness of the article for readers.

Response 5: Thank you very much for your suggestion. We understand that rich visual images are more likely to attract the reader's attention, but due to space constraints we have not been able to show the full range, so we have supplemented this with a selection of smart farming equipment in Figure 1."

Figure 1 has not changed compared to the old version. I still recommend adding illustrations of the use of technical intellectual tools in various sectors of agriculture to the articles.
